# Race, Adolescent Socioeconomic Status, and Lifetime Non-Medical Use of Prescription Painkillers: Evidence from the National Longitudinal Study of Adolescent to Adult Health

**DOI:** 10.3390/ijerph182312289

**Published:** 2021-11-23

**Authors:** Amy Ehntholt, Roman Pabayo, Lisa Berkman, Ichiro Kawachi

**Affiliations:** 1Department of Social and Behavioral Sciences, Harvard T.H. Chan School of Public Health, Boston, MA 02115, USA; lberkman@hsph.harvard.edu (L.B.); ikawachi@hsph.harvard.edu (I.K.); 2School of Public Health, University of Alberta, Edmonton, AB T6G 1C9, Canada; pabayo@ualberta.ca; 3Harvard Center for Population and Development Studies, Cambridge, MA 02138, USA

**Keywords:** health inequities, race, substance use, SES, lifecourse epidemiology

## Abstract

The misuse of prescription painkillers is a major contributor to the ongoing drug overdose epidemic. This study investigated variability in non-medical use of prescription painkillers (NMUPP) by race and early-life socioeconomic status (SES) in a sample now at increased risk for opioid overdose. Data from two waves of the National Longitudinal Study of Adolescent to Adult Health (*n* = 11,602) were used to calculate prevalence of reported NMUPP by Wave 4 (2008; mean age 28), and to assess variation by race and by equivalized household family income at Wave 1 (1994/5). Predicted values for prevalence of NMUPP were modelled, adjusting for age, sex, parental education, and region. Race and SES in adolescence were associated with later reported NMUPP. A gradient was seen in prevalence by SES (adjusted: family income quartile 1 = 13.3%; quartile 2 = 13.8%; quartile 3 = 14.8%; quartile 4 = 16.0%; trend *p*-value = 0.007). Prevalence was higher among males. Racial/ethnic differences in prevalence were seen (non-Hispanic white (NHW) = 18.5%; non-Hispanic black (NHB) = 5.8%; Hispanic = 10.5%; Other = 10.0%). SES differences were less pronounced upon stratification, with trend tests significant only among females (*p* = 0.004), and marginally significant among Hispanic males (*p* = 0.06). Early-life SES was associated with reported lifetime NMUPP: the higher the family income in adolescence, the greater the likelihood of NMUPP by young adulthood. Variations in NMUPP by income paled in comparison with racial/ethnic differences. Results point to a possible long-enduring association between SES and NMUPP, and a need to examine underlying mechanisms.

## 1. Introduction

The ancient Greeks were keenly aware of the potentially dual nature of a drug. Their word for drug, φάρμακον, carries very different meanings based on the context of its appearance: sometimes it signifies a “cure,” other times “poison” [1]. The ongoing epidemic of unintentional drug overdose in the United States is one inextricably tied to the misuse of medicine originally designed to cure pain, but now acting lethally as poison [2].

From 1999 to 2014 the age-adjusted mortality rate due to drug poisoning more than doubled, from 6.1 per 100,000 to 14.7 per 100,000 [3]. By 2011, death rates from motor vehicle crashes had for the first time been surpassed by mortality rates due to drug overdose, making it the new leader in unintentional injury mortality [4]. Between 2004 and 2011 visits to emergency departments attributable to nonmedical use of prescription drugs reportedly increased 132% [5].

Rates of substance use vary by race, with non-Hispanic black (NHB) youth consistently reporting lower lifetime, annual, and recent use of alcohol, cigarettes, and other commonly used drugs compared to their peers in other racial/ethnic groups [6]. Mortality and morbidity rates generally reflect these differences in reported usage. While the age-adjusted drug overdose rate for non-Hispanic whites (NHW) in 2015 was 19.0 per 100,000, for non-Hispanic blacks it was 10.5, and for Hispanics 6.7 [7]. Consistent differences by race and ethnicity in misuse of prescription painkillers have emerged as well. Non-Hispanic white youth in the National Survey on Drug Use and Health (NSDUH) were slightly more likely to report non-medical use of opioids (10.5%) than were Hispanics (9.4%) or non-Hispanic blacks (8.9%) [8]. Similar patterns—with a generally more pronounced protection for blacks—have been seen in recent years [6,9]. Studies involving college students also show higher prevalence of NMUPP among whites than among other groups [10]. One analysis of NSDUH data from 2004-2013 revealed higher self-reported non-medical use of opioids among non-Hispanic whites compared to blacks only until 2013, when no statistically significant difference remained [11].

While studies show that an association between socioeconomic status (SES) and substance use often exists, evidence of its direction remains mixed. Higher SES has been linked to higher and more frequent alcohol consumption [12]. In contrast, a meta-analysis of 10–21-year-olds found evidence of lower SES increasing the likelihood of smoking, but no association between SES and drinking or marijuana use [13]. In one analysis, people of lower income were seen to be more likely to misuse prescription drugs [9]. However, another study using data from the National Household Survey on Drug Abuse (NHSDA) found that individuals with an annual income of USD 20,000–40,000 had a lower likelihood of nonmedical prescription drug use compared to those making more than USD 40,000 per year [14]. Martins and colleagues found higher prevalence of nonmedical prescription drug use among 18-22-year-olds not attending college compared to their peers in university [15]. In a study focusing on current (adult) SES, Stewart and Reed [16] found that recent financial hardship was linked to higher likelihood of non-medical use of prescription drugs; having health insurance was associated with lower rates of misuse. While the relationship between socioeconomic status and treatment of pain (and, therefore, access to painkillers) remains understudied, Joynt and colleagues [17] found evidence of significantly lower rates of prescribing of opioids to patients from lower-SES neighborhoods presenting in emergency departments compared to patients from higher-SES areas.

Existing studies connecting early SES to substance use later in life have offered discrepant findings, and show different associations based on measures used and substances explored. While some research suggests that children from families of lower SES are more likely to smoke [13,18], the association has repeatedly been shown to swing in the opposite direction for alcohol and other substances. Findings have suggested that children from wealthier households are more prone to alcohol abuse and marijuana use when entering adulthood [12,19,20,21]. One review could not determine definite trend in SES and alcohol and marijuana use [13]. These conflicting results highlight the clear need for additional research in this area.

This study investigated the racial, ethnic, and early-life socioeconomic correlates of nonmedical use of prescription painkillers (NMUPP), defined as use without a prescription or use with a prescription but in a manner other than prescribed. It focused on young adults who were teens at the peak of opioid overprescribing—a cohort now at elevated risk of dying of opioid overdose. Although examinations of prescription drug misuse have touched on racial or socioeconomic differences, these studies have often focused solely on the adolescent population [11], or on current SES [16], without delving more deeply into persistent racial differences or effects of family socioeconomic status early in life. Much of the existing evidence relies on cross-sectional data [11]—limiting our understanding of the role earlier factors may play in drug use—or from school-based data, which may mask true differences due to school drop-out rates, themselves often differential by race [6]. Work in lifecourse epidemiology has established the potential far-reaching health-related harms of adversity in early life [22,23]. While past research offers evidence of the link between early SES and other forms of substance use later in life [24,25,26,27,28], there remains a dearth of work investigating the possible effect of family SES in earlier years on misuse of prescription painkillers in adulthood. Therefore, in addition to assessing if prevalence by racial/ethnic group of NMUPP in our sample reflects proportions seen in other data sets, we explored a possible connection between family income while in middle and high school and subsequent report of NMUPP in early adulthood. We hypothesized that significant differences in prescription painkiller misuse would be apparent, with whites reporting more misuse than blacks, and prevalence of misuse by Hispanics falling in between the two. We also hypothesized that we would see differing rates by early family income quartile, with prevalence reported in adulthood increasing with each higher income bracket.

## 2. Materials and Methods

### 2.1. Sample

Data came from the National Longitudinal Study of Adolescent to Adult Health (Add Health), an ongoing, nationally representative sample of students from 132 schools in the U.S. (80 high schools and 52 feeder schools), details of which have been described elsewhere [29]. Systematic sampling methods and implicit stratification in its sample design ensure that the Add Health sample is representative of U.S. schools with respect to region of country, urbanicity, school size, school type, and ethnicity.

Add Health began in 1994, with four instruments used in its first wave (Wave 1, conducted from September of 1994 through December of 1995); two surveys used in Wave 2 (conducted from April through August of 1996); multiple sources in Wave 3 (August 2001 through April of 2002); and one in-home survey conducted in Wave 4 (January of 2008 through February of 2009). Wave 5 data collection was underway as of January 2017. For this investigation, Waves 1 and 4 were used. Instruments included in-school questionnaires, as well as in-home interviews of not only the students themselves, but also their parents and siblings. Other data were collected from school administrators, fellow students, and romantic partners of the students surveyed. Add Health also includes contextual variables garnered from pre-existing databases.

The initial in-school questionnaire was administered to 90,118 students in grades 7 through 12 in the 1994–1995 academic year. Of these 90,118 students, 20,745 were then given more involved at-home surveys and were re-interviewed in the following Waves. Approximately 15,000 participants were interviewed at both Wave 1 and Wave 3, as well as at both Wave 1 and Wave 4. The majority of participants in the survey were aged 18–26 years old at Wave 3, and 24–32 years old at Wave 4. Add Health collects information on respondents’ demographic, social, and behavioral characteristics, along with (more recently) certain biomedical samples.

The sample for this analysis was restricted to those respondents completing in-home surveys at both Waves 1 and 4 (*n* = 15,701). Participants missing parental income data from the parent questionnaire (Wave 1) were excluded (2371 missing and 1328 whose parents refused to report income), as were those missing information on race (*n* = 19). Seventeen additional respondents were dropped for missing outcome, and three more were excluded for missing age data, leaving an analytic sample of 11,602.

### 2.2. Measures

The primary outcome for this study was self-reported lifetime non-medical use of prescription painkillers (NMUPP). In the in-home survey at Wave 4 in 2008, participants were asked a series of questions about substance use, including non-medical use of prescription drugs. The first of these questions assessed lifetime non-medical use of any kind of prescribed drug: “Have you ever taken any prescription drugs that were not prescribed for you, taken prescription drugs in larger amounts than prescribed, more often than prescribed, for longer periods than prescribed, or taken prescription drugs that you took only for the feeling or experience they caused?” Respondents answering “yes” were then instructed to report on specific types of prescription drugs: sedatives, tranquilizers, stimulants, and painkillers. Our binary outcome measure is based on responses to the question capturing painkiller misuse: “Which of the following types of prescription drugs have you taken that were not prescribed for you, taken in larger amounts than prescribed, more often than prescribed, for longer periods than prescribed, or that you took only for the feeling or experience they caused? (check all that apply)—pain killers or opioids, such as Vicodin, OxyContin, Percocet, Demerol, Percodan, or Tylenol with codeine.” For those responding affirmatively, the outcome variable NMUPP was coded “1”; for all others, the variable was coded “0.”

The first of two independent variables of interest came from respondents’ self-reported primary race at Wave 1. Respondents were categorized as non-Hispanic white, non-Hispanic black, Hispanic, and all other races (“other”). The second independent variable captures respondents’ socioeconomic status while in middle or high school through their family’s reported pre-tax income. This measure is derived from parental reports of total pre-tax family income (in USD 1000-increments, top-coded at USD 999,000) given by parents of respondents at Wave I in the separate parent questionnaire administered in 1994. The measure was equivalized for household size by dividing the reported total family income figure by the square root of the number of household members. The sample was then divided into quartiles based on participants’ values for this variable.

### 2.3. Covariates

Both sex and age have been shown to significantly predict non-medical use of prescription drugs [10,30,31,32]. Other covariates of interest include parental educational attainment and region of country (West, Midwest, South, Northeast). Therefore, each of the above variables were included in our adjusted models. Demographic characteristics were taken from data collected at Wave 1. Parental education was measured as the higher of maternal or paternal education level reported in the Wave 1 parental questionnaire and categorized as less than high school; high school; some college; college; or more than college.

Other variables of interest for preliminary descriptive analyses included drug use data that came from answers to in-home survey questions at Wave 4 asking about respondents’ experience with other substances, including any prescription misuse (not limited to painkillers), alcohol, cocaine, marijuana, and methamphetamine. Add Health’s constructed Wave 4 variables were used for the following information on participants in young adulthood: alcohol use or dependence; cannabis use or dependence; other drug use or dependence; smoker status; college graduate status; home ownership; depression; obesity; history of arrest; and rough income measures.

### 2.4. Statistical Analysis

Analyses were conducted using SAS version 9.4 (SAS Institute; Cary, NC, USA) and STATA. Overall crude prevalence of lifetime painkiller misuse was first calculated for the full (unweighted) sample (*n* = 11,602), and then for each of the race categories. The sample was stratified by sex to determine the crude proportion of NMUPP for each subset of race category for both males and females separately. Additional stratification was made by family income quartile at Wave 1 to assess prevalence by early family socioeconomic status. Wilcoxon rank-sum tests were performed to determine whether there existed a statistically significant relationship between NMUPP and parental income quartile for the overall sample, as well as for the separate race and sex categories.

In addition to the assessment of unweighted crude proportion of reported lifetime non-medical prescription painkiller misuse, model-based predicted values of prevalence were calculated after adjustment for the potential confounders.

## 3. Results

### 3.1. Demographics

Table 1 displays demographic characteristics of our Add Health sample (*n* = 11,602).

Mean age of respondents at Wave 1 (1994) was 14 years old, 28 years old at Wave 4 (2008), when our outcome of interest was measured. The majority of the sample (59%; *n* = 6798) was non-Hispanic white. One-fifth (*n* = 2347) of participants were categorized as non-Hispanic black; 15% (*n* = 1722) identified as Hispanic; and 6% (*n* = 735) were categorized as “other.” A little more than half of the sample (52%) was female. Median per person household income (equivalized) at Wave 1 was USD 18,475. Sixty-five percent of the sample had a parent with more than a high school education, and one-third with a parent who had obtained a college degree or higher.

### 3.2. Substance Use

At Wave 4, as Table 1 shows, 14% of the full sample reported lifetime misuse of prescription painkillers; the prevalence of lifetime misuse of any prescription drug was higher, at 17%. Prevalence of misuse of these prescription drugs was higher among non-Hispanic whites than among non-Hispanic blacks: 19% vs. 5%, respectively, for painkillers; 23% vs. 8% for any prescription drugs. These higher proportions of NHW compared to NHB reporting lifetime use were seen for each of the other substances presented in Table 1: for marijuana 60% vs. 45%; cocaine 24% vs. 5%; and crystal meth 12% vs. 2%. Whites were more likely to be classified as ever exhibiting alcohol abuse or dependence at Wave 4 compared to blacks (33% vs. 12%), while numbers for cannabis use or dependence for the two groups were more similar: 14% for NHW and 12% for NHB). Classification of being dependent on or abusing other drugs was more prevalent among whites (10%) than among blacks (3%). Whites were more likely to report being a daily smoker at Wave 4 (27%) than were blacks (16%).

### 3.3. Other Measures at Wave 4

Roughly a third of our sample were college graduates as of the Wave 4 interview, though the percentage was higher for whites (35%) than for blacks (28%). Less than half (41%) reported owning their own homes, with more whites claiming home ownership (48%) than blacks (25%). Reports of “good,” “very good,” or “excellent” self-rated health were very high for the full sample: 91%, and participants meeting criteria for depression represented about 1 in 5 respondents (15% for blacks, 22% for whites). Blacks in our sample were more likely to report having ever been arrested (35%) than were whites (28%) and had higher numbers for obesity (43%) than did whites (34%). Higher median income among whites was roughly USD 35,000, and among blacks just under USD 31,000.

### 3.4. Crude and Adjusted Prevalence of NMUPP

Overall crude prevalence of reported prescription painkiller misuse is presented in Table 2 and Figure 1. The proportion reporting NMUPP in young adulthood was higher among males than among females: 17% versus 12%. A clear gradient appears in the breakdown by income quartile for the full sample, with NMUP reported by 11% of the first quartile, 13% of the second quartile, 16% of the third, and 17% of the fourth. Differential reporting of painkiller misuse is evident by race, with NMUPP apparent among 19% of non-Hispanic white respondents, nearly four times higher than the fraction of non-Hispanic blacks reporting NMUPP (4.9%), and more than twice the percent of Hispanics reporting misuse (8.7%). These differences by race remained after stratification by sex, as did the gradient by income quartile for males alone and for females alone. However, the gradient became less clear when further stratification was done within sex by race. Results of the Wilcoxon rank-sum tests reflect this pattern: only the tests of the trend for the overall sample, all males, all females, all Hispanics, and Hispanic males reveal a statistically significant association between income quartile and misuse of prescription painkillers (respective *p*-values: <0.001, <0.001, <0.001, 0.002, and 0.001).

Model-based predicted values after adjustment for confounders are presented in Table 3 and Figure 2. After controlling for age, sex, parental education, and region of country, the proportions of participants reporting NMUPP changed only slightly (overall: 14.5%), with prevalence across race/ethnicity categories similar to crude prevalence reported above (NHB: 6%; NHW: 19%; Hispanic: 11%; and other: 10%), and with a gradient across income quartiles slightly diminished but still apparent (Q1: 13.3%; Q2: 13.8; Q3: 14.8%; and Q4: 16.0%). Statistically significant trends are seen in these overall numbers, as well as in the pattern for all females. The trend among Hispanic males was found to be marginally significant (*p* = 0.06).

Table 3 notes: Prevalence for sample adjusted for age, sex, parental education reported by parent at Wave 1 (< high school; high school; some college; college; more than college), and region of country (Northeast, Midwest, West, South) using model-based predicted values. ^1^
*p*-values for quartile trend reflect results from extension of (non-parametric) Wilcoxon rank-sum tests (nptrend command in Stata, testing trend for ranks across ordered groups). “NHB” = non-Hispanic black; “NHW” = non-Hispanic white.

### 3.5. Supplemental Analyses

As the models adjusting for confounders revealed a significant increase in prevalence by increasing level of parental education, in supplemental analyses we explored the main effect on NMUPP of parental education, after controlling for parental income, age, sex, and region. Appendix A depicts a somewhat weak gradient by level of parental educational attainment. However, prevalence of NMUPP for those whose parents attained the highest education level of the five categories (>college)—while higher than prevalence for <HS—was consistently lower than the other three categories (HS; <college; college). This pattern was consistent among blacks, whites, and Hispanics, although a noticeable spike was seen in prevalence of NMUPP for those Hispanics whose parents were college graduates.

Appendix A present preliminary results looking at measures of participants’ current SES with respect to their report of lifetime NMUPP in the same year (Wave 4, 2008), adjusting for age, sex, parental education, parental income, and region of the country. Appendix A shows lower prevalence of prescription painkiller misuse among those reporting home ownership, for the sample overall and for each racial group. Appendix A reveals consistently lower rates of lifetime misuse of painkillers among those who were college graduates at Wave 4. Appendix A shows those with higher pre-tax income reporting less misuse of painkillers.

## 4. Discussion

To our knowledge, this is the first study to investigate the variability in lifetime non-medical use of prescription painkillers reported in young adulthood not only by race/ethnicity but also by early-life socioeconomic status (SES)—measured by family household income—as well as by later SES in young adulthood. Our findings support our original hypotheses: prevalence of lifetime NMUPP among non-Hispanic white adults was found to be substantially higher than it was among non-Hispanic blacks and Hispanics, corroborating the evidence of racial disparities seen in previous studies of substance use in general, and prescription drug misuse in particular. In addition, we see a gradient by SES, with higher family income quartiles in early life associated with an increase in reported painkiller misuse in young adulthood. These findings are consistent with previous work suggesting an increased risk of prescription drug misuse with higher socioeconomic status [30,31]. Our results show higher rates of lifetime misuse among males than among females, which contradicts one study’s findings [14], but is similar to gender differences seen in a more recent analysis [16].

Our supplemental analyses identify an intriguing contrast between two relationships: that of prescription painkiller misuse and early-life SES versus NMUPP and current socioeconomic status (as measured by home ownership, college graduate status, and income). While our adjusted models show higher prevalence of painkiller misuse among adults whose childhood SES measures were higher, the supplemental figures indicate that higher SES in adulthood is associated with lower—rather than higher—likelihood of history of painkiller misuse. These findings need not be mutually exclusive. These supplemental results on current SES are not unlike the findings of Stewart and Reed [16], who reported a significant relationship between NMUPP and two of their measures of current adult socioeconomic status: having health insurance (which was found to be protective in adjusted models) and experiencing financial hardship (found to increase the odds of painkiller misuse in their sample). They also echo the findings of Martins and colleagues [15] whose analysis of NSDUH data revealed a protective effect of college attendance on 18–22-year-olds with respect to prescription drug abuse. The relationship seen in our study between NMUPP and early SES (when captured either through parental education or by family household income) might reveal the enduring influence of the consequent access higher SES necessarily brings—to healthcare, and to higher prescription rates. Such exposures in adolescence could establish behavior early on that is carried into adulthood. However, the lack of specificity in the question asked about NMUPP in the Add Health survey prevents us from knowing the exact timing of misuse. It is possible that those with higher family SES in adolescence could therefore more easily access pills to misuse while in school and that such behavior led to poorer health outcomes and also lower SES in young adulthood. On the other hand, those from higher-income families might have also been more likely to receive support for problematic substance use. It is difficult to tease out the nature of these influences given the constraints of our outcome measure. Any association between current lower SES and increased lifetime NMUPP cannot be established as causal, given that both measures were assessed at the same time, and given that the assessment of NMUPP did not ask about timing of misuse. We are unable to determine a direction: it is possible that painkiller misuse might have contributed to participants’ current SES or that the reverse is true, and that their socioeconomic status led to their substance use.

While the mechanisms underlying this consistent variability by race are not fully known, there is strong evidence from both cross-sectional and longitudinal studies suggesting that religiosity has a significant protective influence against the use of cigarettes, alcohol, marijuana, and other drugs [33,34]. Religiosity may also act as a buffer against stress, and guard against substance abuse among teens [35]. Such protection could be more prevalent among blacks and Hispanics, who tend to be more religious [36,37]. A separate protective effect for blacks and Hispanics—for prescription drug misuse in particular—may arise from the well-documented disparities in prescribing by doctors, who in multiple studies have been shown to treat pain differentially by race of patients, with far higher prescribing rates for non-Hispanic whites compared to minorities [38,39,40]. Compounding this disparity is the striking difference in the stocking of opioids in pharmacies, with facilities in minority neighborhoods often understocked, while pharmacies in white neighborhoods carry more adequate quantities of the demanded painkillers [41,42]. An “immigrant effect” might be an alternative explanation for lower rates of substance abuse among Hispanics, who are more likely to be immigrants, a status that has repeatedly been linked to lower rates of cigarette, alcohol, and other illicit drug use [43,44,45,46,47,48]. To our knowledge, neither the “religion hypothesis” nor the “immigrant effect” has specifically been explored in-depth as potential (protective) mechanisms in relation to prescription drug misuse. Further study should include analysis of these measures (available in Add Health).

Unlike other frequently used substances, prescription drugs are at least originally acquired through specific, legal, medical channels. One of the most plausible reasons for such striking differences by race in misuse of painkillers remains the well-established differential prescribing by race, as well as by SES [17]. The evidence of continued disparities in the treatment of pain with opioids analgesic by race is very strong, with, for example, undeniably higher rates of prescribing to white patients visiting emergency departments for pain than to blacks, Hispanics, and Asian and other patients [39,40]. Singhal and colleagues more recently found racial disparities in treatment of pain by emergency room physicians for non-definitive, but not for definitive sources of pain (e.g., backaches rather than fractures), with non-Hispanic blacks up to two-thirds less likely to receive a single dose of opioid in the ER or to be prescribed narcotics [38]. A racially based bias in pain assessment and treatment by medical students and residents was recently exposed, linked to false beliefs about differences in biology between blacks and whites (e.g., that dark skin is thicker, blacks’ blood coagulates more slowly) [49]. All of these combined factors result in a dramatic deficit of opioid prescriptions passing among minority patients, a stark contrast to the abundant supply flowing through their white peers’ networks. Without access to legitimate prescriptions—whether of their own, or of their parents or friends—likelihood of diversion and misuse of such pills by people of color is dramatically decreased.

### Strengths and Limitations

This is the first study to use Add Health data to focus on the possible influence of early-life socioeconomic status (as measured by family household income) and race/ethnicity on lifetime misuse of prescription painkillers reported in young adulthood. This secondary data analysis, following participants for over a decade, takes advantage of a large sample size and self-report of specific types of nonmedical prescription drug use by young adults. We are able to study the previous behavior of adults whose age today puts them in the group most at risk for fatal opioid overdose. Our study brings a much-needed lifecourse perspective to the investigation of risk factors for NMUPP, with its examination of early life exposure. It focuses on specific subsets of the population, filling a noted gap [50], by examining at prevalence by race, and then within race by SES.

While we believe that our study provides valuable new insight, it is not without limitations. First, our measures rely heavily on self-report. While Add Health strives to ensure that participants answer truthfully and accurately by providing more confidential means of answering sensitive questions, there is always a risk of inaccuracy with self-reported information. In our case, both outcome and main independent variables of interest rely on participant and parental self-report. However, it is more likely that substance use would be under-reported rather than over-reported due to desirability bias. Such an error would result in underestimates in our associations, rather than overestimates.

Another limitation of the data lies in the question asked about our outcome, lifetime non-medical use of prescription painkillers. Unfortunately, no questions were asked about the frequency, timing, or extent of misuse. This lack of specificity in timing prevents us from establishing a temporality necessary for establishing causal relationships. Nor were participants queried about the reasons for their misuse of prescription drugs. This lack of detail limits the conclusions we might draw from our results. Questions about misuse of prescription drugs were not asked of participants at baseline (Wave 1, 1994/1995), so we are unable to trace any changes in misuse over time. Additionally, we dropped a substantial number of observations from our sample due to missing income information. Further work should include sensitivity analyses to establish whether including those respondents now excluded from our study would significantly alter our results. Finally, we did not weight our analyses to make our sample nationally representative. Our findings, therefore, cannot be generalized to those beyond our sample population.

## 5. Conclusions

Our results support the hypotheses that early-life SES may have an enduring influence on non-medical use of prescription painkillers that lasts at least into young adulthood, and that misuse varies even more strikingly by race and ethnicity. We also find intriguing evidence of a reversed association between prescription painkiller misuse and SES in adulthood, with higher measures of socioeconomic status tied to lower prevalence of misuse. Our results underscore the need for further research that delves more deeply into the nature of the underlying mechanisms at work, beginning with an investigation of possible effects of religiosity and immigrant status. Our current understanding of the prescription drug problem gripping the nation remains limited and would be vastly enhanced by further research gathering more information on the reasons for misuse, as well as the sources, timing, and frequency of opioid use. A clearer picture of what drives the patterns we see in this analysis should inform policy and education efforts in curbing the ongoing opioid epidemic, in treating those affected, and in altering any detectable upstream causes.

## Figures and Tables

**Figure 1 ijerph-18-12289-f001:**
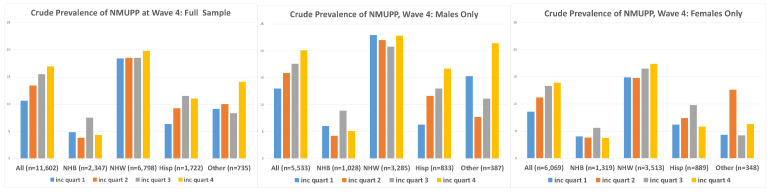
Crude prevalence of NMUPP at Wave 4, by parental income at Wave 1. Note: “W1” = Wave 1; “W4” = Wave 4; “NHB” = non-Hispanic black; “NHW” = non-Hispanic white; “Hisp” = Hispanic.

**Figure 2 ijerph-18-12289-f002:**
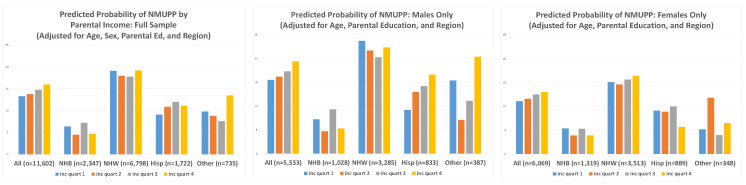
Model-based predicted prevalence of reported NMUPP at Wave 4 (adjusted for age, sex, parental education, and region of country) Note: “NHB” = non-Hispanic black; “NHW” = non-Hispanic white; “Hisp” = Hispanic.

**Table 1 ijerph-18-12289-t001:** Sample characteristics of Add Health (unweighted, *n* = 11,602).

Demographics, W1 (1994/1995)	Mean	%		
Mean Age	14			
Non-Hispanic Black	2347	20%		
Non-Hispanic White	6798	59%		
Hispanic	1722	15%		
Other Race	735	6%		
Male	5533	48%		
Per-person HH Inc. (Equiv.)	USD 18,000 (med.)			
Parental Ed >= College	3858	33%		
Substance Use, W4 (2008)			NHB	NHW
Rx Painkiller Ever Misuse		14%	5.%	19%
Any Rx Drug Ever Misuse		17%	8%	23%
Marijuana Ever		55%	45%	60%
Cocaine Ever		19%	5%	24%
Crystal Meth		9%	2%	12%
Ever Alcohol Use/Dep	3049	26%	12%	33%
Ever Cannabis Use/Dep	1491	13%	12%	14%
Other Drug Use/Dep	880	8%	3%	10%
Daily Smoker	2551	22%	16%	27%
Other Measures, W4 (2008)			NHB	NHW
College Grad	3750	32%	28%	35%
Own Home	4687	41%	25%	48%
Good self-rated health	10,537	91%	90%	92%
Depressed	2215	19%	15%	22%
Obese	4034	37%	43%	34%
Ever Arrested	3336	29%	35%	28%
Midpoint Income >= USD 50K	6082	56%	41%	59%
Income Pre-Tax	USD 35,078 mean; USD 30,000 median		USD 30,743 mean; USD 26,000 med.	USD 35,570 mean; USD 30,000 med.

Notes: “Good self-rated health” is dichotomized as “1” for those responding “good,” “very good,” or “excellent” and “0” for those responding “fair” or “poor” to question asking respondent to describe his or her own health. Eighty respondents are missing data on smoking status; 546 are missing obesity info. “W1” = Wave 1; “W4” = Wave 4; “Per-person HH inc.” = per-person household income; “Rx” = prescription; “NHB” = non-Hispanic black; “NHW” = non-Hispanic white.

**Table 2 ijerph-18-12289-t002:** Crude prevalence (%) of reported NMUPP at Wave 4 (*n* = 11,602).

		Per Person Household Income Quartile (Wave 1)				
	Overall	Q1	Q2	Q3	Q4	*p*-value for trend ^1^
*Full sample (n = 11,602)*	14.1	10.6	13.4	15.5	16.9	<0.001
NHB (*n* = 2347)	4.9	4.8	3.8	7.5	4.3	0.677
NHW (*n* = 6798)	19.0	18.4	18.5	18.5	19.8	0.216
Hispanic (*n* = 1722)	8.7	6.3	9.2	11.5	11.0	0.002
Other (*n* = 735)	10.5	9.1	10.0	8.3	14.1	0.162
*Males only (n = 5533)*	16.7	13.0	15.9	17.6	20.1	<0.001
NHB (*n* = 1028)	5.9	6.0	4.2	8.9	5.1	0.793
NHW (*n* = 3285)	22.1	22.9	22.0	20.8	22.8	0.989
Hispanic (*n* = 833)	10.3	6.3	11.6	13.0	16.7	0.001
Other (*n* = 387)	13.7	15.3	7.7	11.1	21.4	0.101
*Females only (n = 6069)*	11.6	8.6	11.2	13.3	13.9	<0.001
NHB (*n* = 1319)	4.2	4.0	3.8	5.6	3.7	0.780
NHW (*n* = 3513)	16.1	14.9	14.8	16.5	17.4	0.101
Hispanic (*n* = 889)	7.2	6.2	7.4	9.8	5.8	0.503
Other (*n* = 348)	6.9	4.3	12.6	4.2	6.3	0.905

Notes: Crude prevalence does not take into account clustering of students within schools. ^1^
*p*-values for quartile trend reflect results from extension of (non-parametric) Wilcoxon rank-sum tests (nptrend command in Stata, testing trend for ranks across ordered groups). “NHB” = non-Hispanic black; “NHW” = non-Hispanic white.

**Table 3 ijerph-18-12289-t003:** Adjusted prevalence (%) reporting prescription painkiller misuse at Wave 4 (*n* = 11,602).

		Parental Income Quartile (Wave 1)				
	Overall	Q1	Q2	Q3	Q4	*p*-value for trend ^1^
*Full sample (n = 11,602)*	14.5	13.3	13.8	14.8	16.0	0.007
NHB (*n* = 2347)	5.8	6.4	4.5	7.2	4.7	0.407
NHW (*n* = 6798)	18.5	19.1	18.0	17.8	19.2	0.712
Hispanic (*n* = 1722)	10.5	9.1	10.9	12.0	11.1	0.403
Other (*n* = 735)	10.0	9.8	8.8	7.6	13.5	0.343
*Males only (n = 5533)*	17.1	15.5	16.2	17.3	19.4	0.130
NHB (*n* = 1028)	6.7	7.2	4.7	9.3	5.3	0.268
NHW (*n* = 3285)	21.8	23.7	21.7	20.3	22.3	0.493
Hispanic (*n* = 833)	12.3	9.2	13.0	14.2	16.6	0.062
Other (*n* = 387)	13.3	15.4	7.1	11.1	20.4	0.214
*Females only (n = 6069)*	12.1	11.1	11.6	12.5	13.0	0.004
NHB (*n* = 1319)	4.8	5.4	3.9	5.3	3.9	0.996
NHW (*n* = 3513)	15.5	15.1	14.6	15.6	16.4	0.131
Hispanic (*n* = 889)	9.0	9.1	8.9	10.0	5.7	0.361
Other (*n* = 348)	6.9	5.2	11.8	4.0	6.5	0.719

Notes: ^1^
*p*-values for quartile trend reflect results from extension of (non-parametric) Wilcoxon rank-sum tests (nptrend command in Stata, testing trend for ranks across ordered groups).

## Data Availability

The data that support the findings of this study are available from the National Study of Adolescent to Adult Health (Add Health). However, restrictions apply to the availability of these data, which were used under license for the current study, and are not publicly available (these data are “restricted-use”). A smaller, public-use version of these data is available, though this data set would not include all variables used in these analyses. For more information: http://www.cpc.unc.edu/projects/addhealth/documentation/publicdata.

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
