# Peer review of "Race, Adolescent Socioeconomic Status, and Lifetime Non-Medical Use of Prescription Painkillers: Evidence from the National Longitudinal Study of Adolescent to Adult Health"

_ijerph, 2021, doi:10.3390/ijerph182312289_

Round 1

Reviewer 1 Report

The paper concerns an interesting topic. Therefore it's not clear the method used to collect data and in detail what kind of questionnaire the authors use, what answer they did (open/close), and other information about the tools. So the part to be improved should be the description of method.

Author Response

Reply: We thank the reviewer for this feedback. We have made more explicit reference within our manuscript to a paper describing in more detail the Add Health study, data set, and methodology (Harris et al., 2009):

“Data came from the National Longitudinal Study of Adolescent to Adult Health (Add Health), an ongoing, nationally representative sample of students from 132 schools in the U.S. (80 high schools and 52 feeder schools), details of which have been described elsewhere [29].”

We believe this addition strengthens the already fairly extensive details given in the Sample and Measures sections of the Methods (which includes an explanation of the dichotomous nature of the measures, and description of the questions asked):

“2.1 Sample

Data came from the National Longitudinal Study of Adolescent to Adult Health (Add Health), an ongoing, nationally representative sample of students from 132 schools in the U.S. (80 high schools and 52 feeder schools), details of which have been described elsewhere [29]. Systematic sampling methods and implicit stratification in its sample design ensure that the Add Health sample is representative of U.S. schools with respect to region of country, urbanicity, school size, school type, and ethnicity.

Add Health began in 1994, with four instruments used in its first wave (Wave 1, conducted from September of 1994 through December of 1995); two surveys used in Wave 2 (conducted from April through August of 1996); multiple sources in Wave 3 (August 2001 through April of 2002); and one in-home survey conducted in Wave 4 (January of 2008 through February of 2009). Wave 5 data collection was underway as of January 2017. For this investigation, Waves 1 and 4 were used. Instruments included in-school questionnaires, as well as in-home interviews of not only the students themselves, but also their parents and siblings. Other data were collected from school administrators, fellow students, and romantic partners of the students surveyed. Add Health also includes contextual variables garnered from pre-existing databases.

The initial in-school questionnaire was administered to 90,118 students in grades 7 through 12 in the 1994-1995 academic year. Of these 90,118 students, 20,745 were then given more involved at-home surveys, and were re-interviewed in the following Waves. Approximately 15,000 participants were interviewed at both Wave 1 and Wave 3, as well as at both Wave 1 and Wave 4. The majority of participants in the survey were aged 18 to 26 years old at Wave 3, and 24 to 32 years old at Wave 4. Add Health collects information on respondents’ demographic, social, and behavioral characteristics, along with (more recently) certain biomedical samples.

The sample for this analysis was restricted to those respondents completing in-home surveys at both Waves 1 and 4 (n=15,701). Participants missing parental income data from the parent questionnaire (Wave 1) were excluded (2,371 missing and 1,328 whose parents refused to report income), as were those missing information on race (n=19). Seventeen additional respondents were dropped for missing outcome, and three more were excluded for missing age data, leaving an analytic sample of 11,602.

2.2 Measures

The primary outcome for this study was self-reported lifetime non-medical use of prescription painkillers (NMUPP). In the in-home survey at Wave 4 in 2008, participants were asked a series of questions about substance use, including non-medical use of prescription drugs. The first of these questions assessed lifetime non-medical use of any kind of prescribed drug: “Have you ever taken any prescription drugs that were not prescribed for you, taken prescription drugs in larger amounts than prescribed, more often than prescribed, for longer periods than prescribed, or taken prescription drugs that you took only for the feeling or experience they caused?” Respondents answering “yes” were then instructed to report on specific types of prescription drugs: sedatives, tranquilizers, stimulants, and painkillers. Our binary outcome measure is based on responses to the question capturing painkiller misuse: “Which of the following types of prescription drugs have you taken that were not prescribed for you, taken in larger amounts than prescribed, more often than prescribed, for longer periods than prescribed, or that you took only for the feeling or experience they caused? (check all that apply)… pain killers or opioids, such as Vicodin, OxyContin, Percocet, Demerol, Percodan, or Tylenol with codeine.” For those responding affirmatively, the outcome variable NMUPP was coded “1”; for all others, the variable was coded “0.”

The first of two independent variables of interest came from respondents’ self-reported primary race at Wave 1. Respondents were categorized as non-Hispanic white, non-Hispanic black, Hispanic, and all other races (“other”). The second independent variable captures respondents’ socioeconomic status while in middle or high school through their family’s reported pre-tax income. This measure is derived from parental reports of total pre-tax family income (in $1000-increments, top-coded at $999,000) given by parents of respondents at Wave I in the separate parent questionnaire administered in 1994. The measure was equivalized for household size by dividing the reported total family income figure by the square root of the number of household members. The sample was then divided into quartiles based on participants’ values for this variable.”

Reviewer 2 Report

For this manuscript it was investigated if the racial, ethnic, and early-life socioeconomic correlates with nonmedical use of prescription painkillers. The topic is very relevant and the study design and the study procedure are very clear. However, the authors used several abbreviations and they should define them when the name come up in first time.

I would like to make some suggestions for revision:

Abstract 

SES, NHB, NHW- as the authors used these abbreviations, they should define them when they come up in first time.

Introduction

“The ancient Greeks were keenly aware of the potentially dual nature of a drug. Their word for drug, φάρμακον, carries very different meanings based on the context of its appearance: sometimes it signifies a “cure,” other times “poison.” The ongoing epidemic of unintentional drug overdose in the United States is one inextricably tied to the misuse of medicine originally designed to cure pain, but now acting lethally as poison.” Please provide a reference in the end of each phrase.

“This study investigated the racial, ethnic, and early-life socioeconomic correlates of nonmedical use of prescription painkillers (NMUPP), defined as use without a prescription or use with a prescription but in a manner other than prescribed. It focused on young adults who were teens at the peak of opioid overprescribing—a cohort now at elevated risk of dying of opioid overdose.” In my point of view this paragraph should be in the end of the introduction, before of all the hypothesis formulated by the authors.

“Substance use varies by race, with non-Hispanic black youth consistently reporting lower lifetime, annual, and recent use of alcohol, cigarettes, and most other types of drugs”- This phrase is very confuse, please revise it

“However, another study found an apparent protective effect with an income of $40,000 or less [12]”- the authors should contextualize the idea of this phrase.

Table 1- All the abbreviations should be speeled. The authors can put the speeling of each abbreviation as a footnote in the table. Not all readers know the meaning of Rx, for example.

Table 2-

MALES ONLY

(n=5,533) 16.7 13.0 15.9 17.6 20.1 <0.001

NHB (n=1,028) 5.9 6.0 4.2 8.9 5.1 0.793

NHW (n=3,285) 22.1 22.9 22.0 20.8 22.8 0.989

Hispanic (n=833) 10.3 6.3 11.6 13.0 16.7 0.001

Other (n=387) 13.7 15.3 7.7 11.1 21.4 0.101

11.6 8.6 11.2 13.3 13.9

The line marked in the table is not correctly formatted.  The authors should spell the abbreviations in footnote.

Figures 1 and figure 2- The figures have very small subtitles, being difficult to read them.

Table 3- The authors should spell the abbreviations in footnote

Author Response

For this manuscript it was investigated if the racial, ethnic, and early-life socioeconomic correlates with nonmedical use of prescription painkillers. The topic is very relevant and the study design and the study procedure are very clear. However, the authors used several abbreviations and they should define them when the name come up in first time.

I would like to make some suggestions for revision:

Abstract 

SES, NHB, NHW- as the authors used these abbreviations, they should define them when they come up in first time.

  • Reply: We thank the reviewer for this kind response, and for the helpful suggestion. In the abstract we have now fully written out these terms (“socioeconomic status,” “non-Hispanic white,” and “non-Hispanic black”) when they first appear, before abbreviating them. We have done the same within the main text of the paper as well (spelling out first appearances of the abbreviations). We have also clarified in the abstract how SES has been measured in our study.

Introduction

“The ancient Greeks were keenly aware of the potentially dual nature of a drug. Their word for drug, φάρμακον, carries very different meanings based on the context of its appearance: sometimes it signifies a “cure,” other times “poison.” The ongoing epidemic of unintentional drug overdose in the United States is one inextricably tied to the misuse of medicine originally designed to cure pain, but now acting lethally as poison.” Please provide a reference in the end of each phrase.

  • Reply: Thank you for pointing out this omission. We have added relevant references to these two statements:

“The ancient Greeks were keenly aware of the potentially dual nature of a drug. Their word for drug, φάρμακον, carries very different meanings based on the context of its appearance: sometimes it signifies a “cure,” other times “poison” [1]. The ongoing epidemic of unintentional drug overdose in the United States is one inextricably tied to the misuse of medicine originally designed to cure pain, but now acting lethally as poison [2].”

  1. Jacques Derrida, J, Plato's Pharmacy. In Barbara Johnson (ed.), Dissemination.Chicago, IL: University of Chicago Press. pp. 61-171 (1981).
  2. Alexander GC, Frattaroli S, Gielen AC, eds. The Prescription Opioid Epidemic: An Evidence-Based Approach. Johns Hopkins Bloomberg School of Public Health, Baltimore, Maryland: 2015.

“This study investigated the racial, ethnic, and early-life socioeconomic correlates of nonmedical use of prescription painkillers (NMUPP), defined as use without a prescription or use with a prescription but in a manner other than prescribed. It focused on young adults who were teens at the peak of opioid overprescribing—a cohort now at elevated risk of dying of opioid overdose.” In my point of view this paragraph should be in the end of the introduction, before of all the hypothesis formulated by the authors.

  • Reply: We thank the reviewer for this suggestion and agree that this text could be more appropriate placed later in the Introduction. We have moved the paragraph to the start of the last paragraph of the Introduction, preceding our hypothesis, as recommended.

“Substance use varies by race, with non-Hispanic black youth consistently reporting lower lifetime, annual, and recent use of alcohol, cigarettes, and most other types of drugs”- This phrase is very confuse, please revise it

  • Reply: Upon reflection we agree that this sentence was rather confusing. We have rephrased it:

“Rates of substance use vary by race, with non-Hispanic black (NHB) youth consistently reporting lower lifetime, annual, and recent use of alcohol, cigarettes, and other commonly used drugs compared to their peers in other racial/ethnic groups [4].”

“However, another study found an apparent protective effect with an income of $40,000 or less [12]”- the authors should contextualize the idea of this phrase.

  • Reply: We appreciate this suggestion. We have revised the sentence as follows:

“However, another study using data from the National Household Survey on Drug Abuse (NHSDA) found that individuals with an annual income of $20,000-$40,000 had a lower likelihood of nonmedical prescription drug use compared to those making more than $40,000 per year [14].”

Table 1- All the abbreviations should be spelled. The authors can put the spelling of each abbreviation as a footnote in the table. Not all readers know the meaning of Rx, for example.

  • Reply: Thank you for this suggestion. We have added explanations of abbreviations to the Notes for Table 1.

Table 2-

MALES ONLY

(n=5,533) 16.7 13.0 15.9 17.6 20.1 <0.001

NHB (n=1,028) 5.9 6.0 4.2 8.9 5.1 0.793

NHW (n=3,285) 22.1 22.9 22.0 20.8 22.8 0.989

Hispanic (n=833) 10.3 6.3 11.6 13.0 16.7 0.001

Other (n=387) 13.7 15.3 7.7 11.1 21.4 0.101

11.6 8.6 11.2 13.3 13.9

The line marked in the table is not correctly formatted.  The authors should spell the abbreviations in footnote.

  • Reply: We believe there may have been a formatting issue during submission, since we do not see an incorrect formatting of the line in Table 2 for Males Only.

We have added the appropriate explanations for the race/ethnicity abbreviations to the Table 2 Notes.

Figures 1 and figure 2- The figures have very small subtitles, being difficult to read them.

  • Reply: We see what the reviewer means. We have further enlarged the subtitles as much as possible (given space constraints).

Table 3- The authors should spell the abbreviations in footnote

  • Reply: We have added abbreviations in Notes for both Figures 1 and 2.

Round 2

Reviewer 2 Report

The authors clearly improved the manuscript, therefore, in my opinion, it is now susceptible for publication